# Learning Stochastic Feedforward Neural Networks

**Yichuan Tang**
Department of Computer Science
University of Toronto
Toronto, Ontario, Canada.
tang@cs.toronto.edu

**Ruslan Salakhutdinov**
Department of Computer Science and Statistics
University of Toronto
Toronto, Ontario, Canada.
rsalakhu@cs.toronto.edu

## Abstract

Multilayer perceptrons (MLPs) or neural networks are popular models used for nonlinear regression and classification tasks. As regressors, MLPs model the conditional distribution of the predictor variables $Y$ given the input variables $X$. However, this predictive distribution is assumed to be unimodal (e.g. Gaussian). For tasks involving structured prediction, the conditional distribution should be multi-modal, resulting in one-to-many mappings. By using stochastic hidden variables rather than deterministic ones, Sigmoid Belief Nets (SBNs) can induce a rich multimodal distribution in the output space. However, previously proposed learning algorithms for SBNs are not efficient and unsuitable for modeling real-valued data. In this paper, we propose a stochastic feedforward network with hidden layers composed of *both deterministic and stochastic* variables. A new Generalized EM training procedure using importance sampling allows us to efficiently learn complicated conditional distributions. Our model achieves superior performance on synthetic and facial expressions datasets compared to conditional Restricted Boltzmann Machines and Mixture Density Networks. In addition, the latent features of our model improves classification and can learn to generate colorful textures of objects.

## 1 Introduction

Multilayer perceptrons (MLPs) are general purpose function approximators. The outputs of a MLP can be interpreted as the sufficient statistics of a member of the exponential family (conditioned on the input $X$), thereby inducing a distribution over the output space $Y$. Since the nonlinear activations are all *deterministic*, MLPs model the conditional distribution $p(Y|X)$ with a unimodal assumption (e.g. an isotropic Gaussian)[1].

For many structured prediction problems, we are interested in a conditional distribution $p(Y|X)$ that is multimodal and may have complicated structure[2]. One way to model the multi-modality is to make the hidden variables stochastic. Conditioned on a particular input $X$, different hidden configurations lead to different $Y$. Sigmoid Belief Nets (SBNs) [3, 2] are models capable of satisfying the multi-modality requirement. With binary input, hidden, and output variables, they can be viewed as directed graphical models where the sigmoid function is used to compute the degrees of "belief" of a child variable given the parent nodes. Inference in such models is generally intractable. The original paper by Neal [2] proposed a Gibbs sampler which cycles through the hidden nodes one at a time. This is problematic as Gibbs sampling can be very slow when learning large models or fitting moderately-sized datasets. In addition, slow mixing of the Gibbs chain would typically lead to a biased estimation of gradients during learning. A variational learning algorithm based on

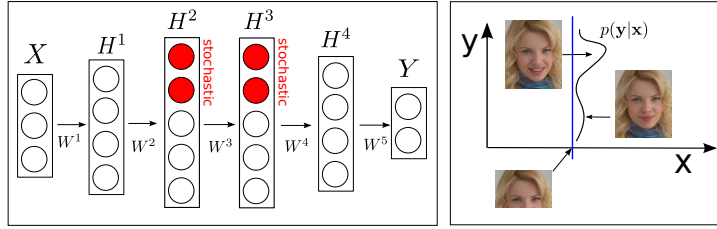

Figure 1: *Stochastic Feedforward Neural Networks.* **Left***: Network diagram. Red nodes are stochastic and binary, while the rest of the hiddens are deterministic sigmoid nodes.* **Right***: motivation as to why multimodal outputs are needed. Given the top half of the face* **x***, the mouth in* **y** *can be different, leading to different expressions.*

the mean-field approximation was proposed in [4] to improve the learning of SBNs. A drawback of the variational approach is that, similar to Gibbs, it has to cycle through the hidden nodes one at a time. Moreover, beside the standard mean-field variational parameters, additional parameters must be introduced to lower-bound an intractable term that shows up in the expected free energy, making the lower-bound looser. Gaussian fields are used in [5] for inference by making Gaussian approximations to units' input, but there is no longer a lower bound on the likelihood.

In this paper, we introduce the Stochastic Feedforward Neural Network (SFNN) for modeling conditional distributions $p(\mathbf{y}|\mathbf{x})$ over *continuous* real-valued $Y$ output space. Unlike SBNs, to better model continuous data, SFNNs have hidden layers with *both* stochastic *and* deterministic units. The left panel of Fig. 1 shows a diagram of SFNNs with multiple hidden layers. Given an input vector $\mathbf{x}$, different states of the stochastic units can generates different modes in $Y$. For learning, we present a novel Monte Carlo variant of the Generalized Expectation Maximization algorithm. Importance sampling is used for the E-step for inference, while error backpropagation is used by the M-step to improve a variational lower bound on the data log-likelihood. SFNNs have several attractive properties, including:

- We can draw samples from the exact model distribution without resorting to MCMC.
- Stochastic units form a distributed code to represent an exponential number of mixture components in output space.
- As a directed model, learning does not need to deal with a global partition function.
- Combination of stochastic and deterministic hidden units can be jointly trained using the backpropagation algorithm, as in standard feed-forward neural networks.

The two main alternative models are Conditional Gaussian Restricted Boltzmann Machines (C-GRBMs) [6] and Mixture Density Networks (MDNs) [1]. Note that Gaussian Processes [7] and Gaussian Random Fields [8] are unimodal and therefore incapable of modeling a multimodal $Y$. Conditional Random Fields [9] are widely used in NLP and vision, but often assume $Y$ to be discrete rather than continuous. C-GRBMs are popular models used for human motion modeling [6], structured prediction [10], and as a higher-order potential in image segmentation [11]. While C-GRBMs have the advantage of exact inference, they are energy based models that define *different* partition functions for different input $X$. Learning also requires Gibbs sampling which is prone to poor mixing. MDNs use a mixture of Gaussians to represent the output $Y$. The components' means, mixing proportions, and the output variances are all predicted by a MLP conditioned on $X$. As with SFNNs, the backpropagation algorithm can be used to train MDNs efficiently. However, the number of mixture components in the output $Y$ space must be pre-specified and the number of parameters is linear in the number of mixture components. In contrast, with $N_h$ stochastic hidden nodes, SFNNs can use its distributed representation to model up to $2^{N_h}$ mixture components in the output $Y$.

## 2  Stochastic Feedforward Neural Networks

SFNNs contain binary stochastic hidden variables $\mathbf{h} \in \{0,1\}^{N_h}$, where $N_h$ is the number of hidden nodes. For clarity of presentation, we construct a SFNN from a one-hidden-layer MLP by replacing the sigmoid nodes with stochastic binary ones. Note that other types stochastic units can also be used. The conditional distribution of interest, $p(y|\mathbf{x})$, is obtained by marginalizing out the latent stochastic hidden variables: $p(y|\mathbf{x}) = \sum_{\mathbf{h}} p(y, \mathbf{h}|\mathbf{x})$. SFNNs are directed graphical models where the generative process starts from $\mathbf{x}$, flows through $\mathbf{h}$, and then generates output $y$. Thus, we can factorize the joint distribution as: $p(y, \mathbf{h}|\mathbf{x}) = p(y|\mathbf{h})p(\mathbf{h}|\mathbf{x})$. To model real-valued $y$, we have

$p(y|\mathbf{h}) = \mathcal{N}(y|W_2\mathbf{h} + b_2, \sigma_y^2)$ and $p(\mathbf{h}|\mathbf{x}) = \sigma(W_1\mathbf{x} + b_1)$, where $b$ is the bias. Since $\mathbf{h} \in \{0, 1\}^{N_h}$ is a vector of Bernoulli random variables, $p(y|\mathbf{x})$ has *potentially* $2^{N_h}$ different modes[3], one for every possible binary configurations of $\mathbf{h}$. The fact that $\mathbf{h}$ can take on different states in SFNN is the reason why we can learn one-to-many mappings, which would be impossible with standard MLPs.

The modeling flexibility of SFNN comes with computational costs. Since we have a mixture model with potentially $2^{N_h}$ components conditioned on any $\mathbf{x}$, $p(y|\mathbf{x})$ does not have a closed-form expression. We can use Monte Carlo approximation with $M$ samples for its estimation:

$$p(y|\mathbf{x}) \simeq \frac{1}{M} \sum_{m=1}^{M} p(y|\mathbf{h}^{(m)}), \qquad \mathbf{h}^{(m)} \sim p(\mathbf{h}|\mathbf{x}). \qquad (1)$$

This estimator is unbiased and has relatively low variance. This is because the accuracy of the estimator does not depend on the dimensionality of $\mathbf{h}$ and that $p(\mathbf{h}|\mathbf{x})$ is factorial, meaning that we can draw samples from the *exact* distribution.

If $\mathbf{y}$ is discrete, it is sufficient for all of the hiddens to be discrete. However, using only discrete hiddens is suboptimal when modeling real-valued output $Y$. This is due to the fact that while $y$ is continuous, there are only a finite number of discrete hidden states, each one (e.g. $\mathbf{h}'$) leads to a component which is a Gaussian: $p(y|\mathbf{h}') = \mathcal{N}(y|\mu(\mathbf{h}'), \sigma_y^2)$. The mean of a Gaussian component is a function of the hidden state: $\mu(\mathbf{h}') = W_2^\mathsf{T}\mathbf{h}' + b_2$. When $\mathbf{x}$ varies, only the probability of *choosing* a specific hidden state $\mathbf{h}'$ changes via $p(\mathbf{h}'|\mathbf{x})$, not $\mu(\mathbf{h}')$. However, if we allow $\mu(\mathbf{h}')$ to be a deterministic function of $\mathbf{x}$ as well, we can learn a smoother $p(y|\mathbf{x})$, even when it is desirable to learn small residual variances $\sigma_y^2$. This can be accomplished by allowing for both stochastic *and* deterministic units in a single SFNN hidden layer, allowing the mean $\mu(\mathbf{h}', \mathbf{x})$ to have contributions from two components, one from the hidden state $\mathbf{h}'$, and another one from defining a deterministic mapping from $\mathbf{x}$. As we demonstrate in our experimental results, this is crucial for learning good density models of the real-valued $Y$.

In SFNNs with only one hidden layer, $p(\mathbf{h}|\mathbf{x})$ is a factorial Bernoulli distribution. If $p(\mathbf{h}|\mathbf{x})$ has low entropy, only a few discrete $\mathbf{h}$ states out of the $2^{N_h}$ total states would have any significant probability mass. We can increase the entropy over the stochastic hidden variables by adding a second hidden layer. The second hidden layer takes the stochastic and any deterministic hidden nodes of the first layer as its input. This leads to our proposed SFNN model, shown in Fig. 1.

In our SFNNs, we assume a conditional diagonal Gaussian distribution for the output $Y$: $\log p(\mathbf{y}|\mathbf{h}, \mathbf{x}) \propto -\frac{1}{2} \sum_i \log \sigma_i^2 - \frac{1}{2} \sum_i (y_i - \mu(\mathbf{h}, \mathbf{x}))^2/\sigma_i^2$. We note that we can also use any other parameterized distribution (e.g. Student's t) for the output variables. This is a win compared to the Boltzmann Machine family of models, which require the output distribution to be from the exponential family.

## 2.1 Learning

We present a Monte Carlo variant of the Generalized EM algorithm [12] for learning SFNNs. Specifically, importance sampling is used during the E-step to approximate the posterior $p(\mathbf{h}|y, \mathbf{x})$, while the Backprop algorithm is used during the M-step to calculate the derivatives of the parameters of both the stochastic and deterministic nodes. Gradient ascent using the derivatives will guarantee that the variational lower bound of the model log-likelihood will be improved. The drawback of our learning algorithm is the requirement of sampling the stochastic nodes $M$ times for every weight update. However, as we will show in the experimental results, 20 samples is sufficient for learning good SFNNs.

The requirement of sampling is typical for models capable of structured learning. As a comparison, energy based models, such as conditional Restricted Boltzmann Machines, require MCMC sampling per weight update to estimate the gradient of the log-partition function. These MCMC samples do not converge to the true distribution, resulting in a biased estimate of the gradient.

For clarity, we provide the following derivations for SFNNs with one hidden layer containing only stochastic nodes[4]. For any approximating distribution $q(\mathbf{h})$, we can write down the following varia-

tional lower-bound on the data log-likelihood:

$$\log p(y|\mathbf{x}) = \log \sum_{\mathbf{h}} p(y, \mathbf{h}|\mathbf{x}) = \sum_{\mathbf{h}} p(\mathbf{h}|y, \mathbf{x}) \log \frac{p(y, \mathbf{h}|\mathbf{x})}{p(\mathbf{h}|y, \mathbf{x})} \geq \sum_{\mathbf{h}} q(\mathbf{h}) \log \frac{p(y, \mathbf{h}|\mathbf{x}; \theta)}{q(\mathbf{h})}, \quad (2)$$

where $q(\mathbf{h})$ can be any arbitrary distribution. For the tightest lower-bound, $q(\mathbf{h})$ need to be the exact posterior $p(\mathbf{h}|y, \mathbf{x})$. While the posterior $p(\mathbf{h}|y, \mathbf{x})$ is hard to compute, the "conditional prior" $p(\mathbf{h}|\mathbf{x})$ is easy (corresponds to a simple feedforward pass). We can therefore set $q(\mathbf{h}) \triangleq p(\mathbf{h}|\mathbf{x})$. However, this would be a very bad approximation as learning proceeds, since the learning of the likelihood $p(y|\mathbf{h}, \mathbf{x})$ will increase the KL divergence between the conditional prior and the posterior. Instead, it is critical to use importance sampling with the conditional prior as the proposal distribution.

Let $Q$ be the expected complete data log-likelihood, which is a lower bound on the log-likelihood that we wish to maximize:

$$Q(\theta, \theta_{old}) = \sum_{\mathbf{h}} \frac{p(\mathbf{h}|y, \mathbf{x}; \theta_{old})}{p(\mathbf{h}|\mathbf{x}; \theta_{old})} p(\mathbf{h}|\mathbf{x}; \theta_{old}) \log p(y, \mathbf{h}|\mathbf{x}; \theta) \simeq \frac{1}{M} \sum_{m=1}^{M} w^{(m)} \log p(y, \mathbf{h}^{(m)}|\mathbf{x}; \theta),$$
(3)

where $\mathbf{h}^{(m)} \sim p(\mathbf{h}|\mathbf{x}; \theta_{old})$ and $w^{(m)}$ is the importance weight of the $m$-th sample from the proposal distribution $p(\mathbf{h}|\mathbf{x}; \theta_{old})$. Using Bayes Theorem, we have

$$w^{(m)} = \frac{p(\mathbf{h}^{(m)}|y, \mathbf{x}; \theta_{old})}{p(\mathbf{h}^{(m)}|\mathbf{x}; \theta_{old})} = \frac{p(y|\mathbf{h}^{(m)}, \mathbf{x}; \theta_{old})}{p(y|\mathbf{x}; \theta_{old})} \simeq \frac{p(y|\mathbf{h}^{(m)}; \theta_{old})}{\frac{1}{M} \sum_{m=1}^{M} p(y|\mathbf{h}^{(m)}; \theta_{old})}. \quad (4)$$

Eq. 1 is used to approximate $p(y|\mathbf{x}; \theta_{old})$. For convenience, we define the partial objective of the $m$-th sample as $Q^{(m)} \triangleq w^{(m)} \big( \log p(y|\mathbf{h}^{(m)}; \theta) + \log p(\mathbf{h}^{(m)}|\mathbf{x}; \theta) \big)$. We can then approximate our objective function $Q(\theta, \theta_{old})$ with $M$ samples from the proposal: $Q(\theta, \theta_{old}) \simeq \frac{1}{M} \sum_{m=1}^{M} Q^{(m)}(\theta, \theta_{old})$. For our generalized M-step, we seek to perform gradient ascent on $Q$:

$$\frac{\partial Q}{\partial \theta} \simeq \frac{1}{M} \sum_{m=1}^{M} \frac{\partial Q^{(m)}(\theta, \theta_{old})}{\partial \theta} = \frac{1}{M} \sum_{m=1}^{M} w^{(m)} \frac{\partial}{\partial \theta} \Big\{ \log p(y|\mathbf{h}^{(m)}; \theta) + \log p(\mathbf{h}^{(m)}|\mathbf{x}; \theta) \Big\}. \quad (5)$$

The gradient term $\frac{\partial}{\partial \theta} \{ \cdot \}$ is computed using error backpropagation of two sub-terms. The first part, $\frac{\partial}{\partial \theta} \{ \log p(y|\mathbf{h}^{(m)}; \theta) \}$, treats $y$ as the targets and $\mathbf{h}^{(m)}$ as the input data, while the second part, $\frac{\partial}{\partial \theta} \{ \log p(\mathbf{h}^{(m)}|\mathbf{x}; \theta) \}$, treats $\mathbf{h}^{(m)}$ as the targets and $\mathbf{x}$ as the input data. In SFNNs with a mixture of deterministic and stochastic units, backprop will additionally propagate error information from the first part to the second part.

The full gradient is a weighted summation of the $M$ partial derivatives, where the weighting comes from how well a particular state $\mathbf{h}^{(m)}$ can generate the data $y$. This is intuitively appealing, since learning adjusts both the "preferred" states' abilities to generate the data (first part in the braces), as well as increase their probability of being picked conditioning on $\mathbf{x}$ (second part in the braces). The detailed EM learning algorithm for SFNNs is listed in Alg. 1 of the Supplementary Materials.

## 2.2 Cooperation during learning

We note that for importance sampling to work well in general, a key requirement is that the proposal distribution is not small where the true distribution has significant mass. However, things are slightly different when using importance sampling during learning. Our proposal distribution $p(\mathbf{h}|\mathbf{x})$ and the posterior $p(\mathbf{h}|y, \mathbf{x})$ are not fixed but rather governed by the model parameters. Learning adapts these distribution in a synergistic and cooperative fashion.

Let us hypothesize that at a particular learning iteration, the conditional prior $p(\mathbf{h}|\mathbf{x})$ is small in certain regions where $p(\mathbf{h}|y, \mathbf{x})$ is large, which is undesirable for importance sampling. The E-step will draw $M$ samples and weight them according to Eq. 4. While all samples $\mathbf{h}^{(m)}$ will have very low log-likelihood due to the bad conditional prior, there will be a certain preferred state $\hat{\mathbf{h}}$ with the largest weight. Learning using Eq. 5 will accomplish two things: (1) it will adjust the generative weights to allow preferred states to better generate the observed $y$; (2) it will make the conditional prior better by making it more likely to predict $\hat{\mathbf{h}}$ given $\mathbf{x}$. Since

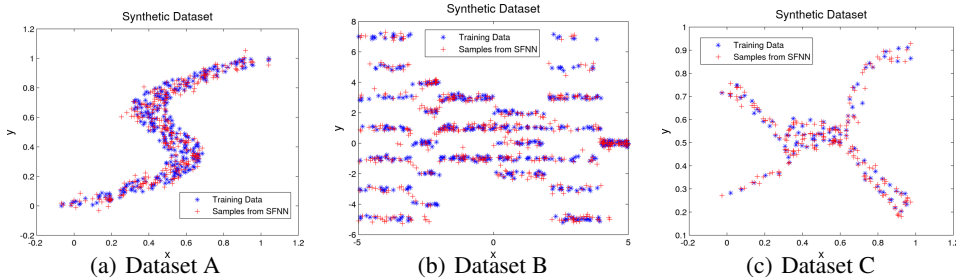

| (a) Dataset A | (b) Dataset B | (c) Dataset C |

Figure 3: *Three synthetic datasets of 1-dimensional one-to-many mappings. For any given $x$, multiple modes in $y$ exist. Blue stars are the training data, red pluses are* exact *samples from SFNNs. Best viewed in color.*

the generative weights are shared, the fact that $\hat{\mathbf{h}}$ generates $y$ accurately will probably reduce the likelihood of $y$ under another state $\tilde{\mathbf{h}}$. The updated conditional prior tends to be a better proposal distribution for the updated model. The cooperative interaction between the conditional prior and posterior during learning provides some robustness to the importance sampler.

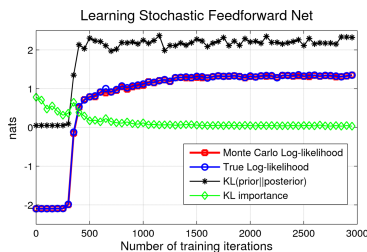

Figure 2: *KL divergence and log-likelihoods. Best viewed in color.*

Empirically, we can see this effect as learning progress on Dataset A of Sec. 3.1 in Fig. 2. The plot shows the model log-likelihood given the training data as learning progresses until 3000 weight updates. 30 importance samples are used during learning with 2 hidden layers of 5 stochastic nodes. We chose 5 nodes because it is small enough that the true log-likelihood can be computed using brute-force integration. As learning progresses, the Monte Carlo approximation is very close to the true log-likelihood using only 30 samples. As expected, the KL from the posterior and prior diverges as the generative weights better models the multi-modalities around $x = 0.5$. We also compared the KL divergence between our empirical weighted importance sampled distribution and true posterior, which converges toward zero. This demonstrate that the prior distribution have learned to *not* be small in regions of large posterior. In other words, this shows that the E-step in the learning of SFNNs is close to exact for this dataset and model.

## 3   Experiments

We first demonstrate the effectiveness of SFNN on synthetic one dimensional one-to-many mapping data. We then use SFNNs to model face images with varying facial expressions and emotions. SFNNs outperform other competing density models by a large margin. We also demonstrate the usefulness of latent features learned by SFNNs for expression classification. Finally, we train SFNNs on a dataset with in-depth head rotations, a database with colored objects, and a image segmentation database. By drawing samples from these trained SFNNs, we obtain qualitative results and insights into the modeling capacity of SFNNs. We provide computation times for learning in the Supplementary Materials.

### 3.1   Synthetic datasets

As a proof of concept, we used three one dimensional one-to-many mapping datasets, shown in Fig. 3. Our goal is to model $p(y|x)$. Dataset A was used by [1] to evaluate the performance of the Mixture Density Networks (MDNs). Dataset B has a large number of tight modes conditioned on any given x, which is useful for testing a model's ability to learn many modes and a small residual variance. Dataset C is used for testing whether a model can learn modes that are far apart from each other. We randomly split the data into a training, validation, and a test set. We report the average test set log-probability averaged over 5 folds for different models in Table 1. The method called 'Gaussian' is a 2D Gaussian estimated on $(x, y)$ jointly, and we report $\log p(y|x)$ which can be obtained easily in closed-form. For Conditional Gaussian Restricted Boltzmann Machine (C-GRBM) we used 25-step Contrastive Divergence [13] (CD-25) to estimate the gradient of the log partition function. We used Annealed Importance Sampling [14, 15] with 50,000 intermediate temperatures to estimate the partition function. SBN is a Sigmoid Belief Net with three hidden stochastic binary layers between the input and the output layer. It is trained in the same way as SFNN, but there are no deterministic units. Finally, SFNN has four hidden layers with the inner

two being hybrid stochastic/deterministic layers (See Fig. 1). We used 30 importance samples to approximate the posterior during the E-step. All other hyper-parameters for all of the models were chosen to maximize the validation performance.

| | Gaussian | MDN | C-GRBM | SBN | SFNN |
|---|---|---|---|---|---|
| A | 0.078±0.02 | 1.05±0.02 | 0.57±0.01 | 0.79±0.03 | 1.04±0.03 |
| B | -2.40±0.07 | -1.58±0.11 | -2.14±0.04 | -1.33±0.10 | -0.98±0.06 |
| C | 0.37±0.07 | 2.03±0.05 | 1.36±0.05 | 1.74±0.08 | 2.21±0.16 |

Table 1: *Average test log-probability density on synthetic 1D datasets.*

Table 1 reveals that SFNNs consistently outperform all other methods. Fig. 3 further shows samples drawn from SFNNs as red 'pluses'. Note that SFNNs can learn small residual variances to accurately model Dataset B. Comparing SBNs to SFNNs, it is clear that having deterministic hidden nodes is a big win for modeling continuous $y$.

## 3.2    Modeling Facial Expression

Conditioned on a subject's face with neutral expression, the distribution of all possible emotions or expressions of this particular individual is multimodal in pixel space. We learn SFNNs to model facial expressions in the Toronto Face Database [16]. The Toronto Face Database consist of 4000 images of 900 individuals with 7 different expressions. Of the 900 subjects, there are 124 with 10 or more images per subject, which we used as our data. We randomly selected 100 subjects with 1385 total images for training, while 24 subjects with a total of 344 images were selected as the test set.

For each subject, we take the average of their face images as $\mathbf{x}$ (mean face), and learn to model this subject's varying expressions $\mathbf{y}$. Both $\mathbf{x}$ and $\mathbf{y}$ are grayscale and downsampled to a resolution of $48 \times 48$. We trained a SFNN with 4 hidden layers of size 128 on these facial expression images. The second and third "hybrid" hidden layers contained 32 stochastic binary and 96 deterministic hidden nodes, while the first and the fourth hidden layers consisted of only deterministic sigmoids. We refer to this model as SFNN2. We also tested the same model but with only one hybrid hidden layer, that we call SFNN1. We used mini-batches of size 100 and and 30 importance samples for the E-step. A total of 2500 weight updates were performed. Weights were randomly initialized with standard deviation of 0.1, and the residual variance $\sigma_y^2$ was initialized to the variance of $\mathbf{y}$.

For comparisons with other models, we trained a Mixture of Factor Analyzers (MFA) [17], Mixture Density Networks (MDN), and Conditional Gaussian Restricted Boltzmann Machines (C-GRBM) on this task. For the Mixture of Factor Analyzers model, we trained a mixture with 100 components, one for each training individual. Given a new test face $\mathbf{x}_{test}$, we first find the training $\hat{\mathbf{x}}$ which is closest in Euclidean distance. We then take the parameters of the $\hat{\mathbf{x}}$'s FA component, while replacing the FA's mean with $\mathbf{x}_{test}$. Mixture Density Networks is trained using code provided by the NETLAB package [18]. The number of Gaussian mixture components and the number of hidden nodes were selected using a validation set. Optimization is performed using the scaled conjugate gradient algorithm until convergence. For C-GRBMs, we used CD-25 for training. The optimal number of hidden units, selected via validation, was 1024. A population sparsity objective on the hidden activations was also part of the objective [19]. The residual diagonal covariance matrix is also learned. Optimization used stochastic gradient descent with mini-batches of 100 samples each.

| | MFA | MDN | C-GRBM | SFNN1 | SFNN2 |
|---|---|---|---|---|---|
| Nats | 1406±52 | 1321±16 | 1146±113 | 1488±18 | 1534±27 |
| Time | 10 secs. | 6 mins. | 158 mins. | 112 secs. | 113 secs. |

Table 2: *Average test log-probability and total training time on facial expression images. Note that for continuous data, these are probability densities and can be positive.*

Table 2 displays the average log-probabilities along with standard errors of the 344 test images. We also recorded the total training time of each algorithm, although this depends on the number of weight updates and whether or not GPUs are used (see the Supplementary Materials for more details). For MFA and MDN, the log-probabilities were computed exactly. For SFNNs, we used Eq. 1 with 1000 samples. We can see that SFNNs substantially outperform all other models. Having two hybrid hidden layers (SFNN2) improves model performance over SFNN1, which has only one hybrid hidden layer.

Qualitatively, Fig. 4 shows samples drawn from the trained models. The leftmost column are the mean faces of 3 test subjects, followed by 7 samples from the distribution $p(\mathbf{y}|\mathbf{x})$. For C-GRBM, samples are generated from a Gibbs chain, where each successive image is taken after 1000 steps. For the other 2 models, displayed samples are *exact*. MFAs overfit on the training set, generating

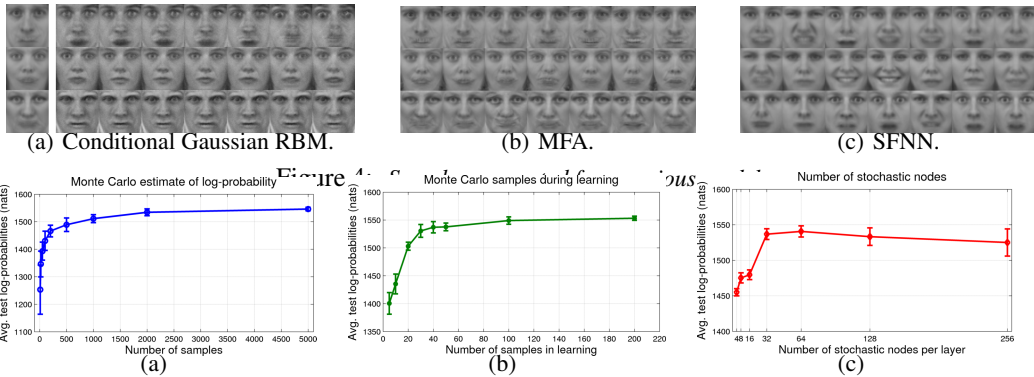

(a) Conditional Gaussian RBM.      (b) MFA.      (c) SFNN.

Figure 5: *Plots demonstrate how hyperparameters affect the evaluation and learning of SFNNs.*

samples with significant artifacts. Samples produced by C-GRBMs suffer from poor mixing and get stuck at a local mode. SFNN samples show that the model was able to capture a combination of mutli-modality and preserved much of the identity of the test subjects. We also note that SFNN generated faces are not simple memorization of the training data. This is validated by its superior performance on the *test* set in Table 2.

We further explored how different hyperparameters (e.g. # of stochastic layers, # of Monte Carlo samples) can affect the learning and evaluation of SFNNs. We used face images and SFNN2 for these experiments. First, we wanted to know the number of $M$ in Eq. 1 needed to give a reasonable estimate of the log-probabilities. Fig. 5(a) shows the estimates of the log-probability as a function of the number of samples. We can see that having about 500 samples is reasonable, but more samples provides a slightly better estimate. The general shape of the plot is similar for all other datasets and SFNN models. When $M$ is small, we typically underestimate the true log-probabilities. While 500 or more samples are needed for accurate model *evaluation*, only 20 or 30 samples are sufficient for learning good models (as shown in Fig. 5(b). This is because while $M = 20$ gives suboptimal approximation to the true posterior, learning still improves the variational lower-bound. In fact, we can see that the difference between using 30 and 200 samples during learning results in only about 20 nats of the final average test log-probability. In Fig. 5(c), we varied the number of binary stochastic hidden variables in the 2 inner hybrid layers. We did not observe significant improvements beyond more than 32 nodes. With more hidden nodes, over-fitting can also be a problem.

### 3.2.1 Expression Classification

The internal hidden representations learned by SFNNs are also useful for classification of facial expressions. For each $\{\mathbf{x}, \mathbf{y}\}$ image pair, there are 7 possible expression types: neutral, angry, happy, sad, surprised, fear, and disgust. As baselines, we used regularized linear softmax classifiers and multilayer perceptron classifier taking pixels as input. The mean of every pixel across all cases was set to 0 and standard deviation was set to 1.0. We then append the learned hidden features of SFNNs and C-GRBMs to the image pixels and re-train the same classifiers. The results are shown in the first row of Table 3. Adding hidden features from the SFNN trained in an unsupervised manner (without expression labels) improves accuracy for both linear and nonlinear classifiers.

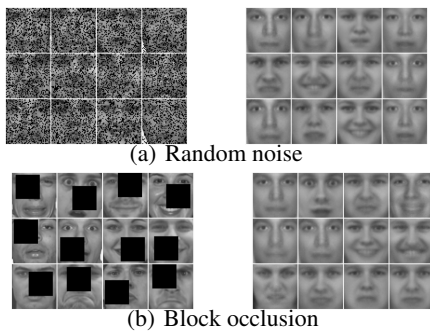

(a) Random noise

(b) Block occlusion

Figure 6: **Left**: *Noisy test images* $\mathbf{y}$. *Posterior inference in SFNN finds* $E_{p(\mathbf{h}|\mathbf{x},\mathbf{y})}[\mathbf{h}]$. **Right**: *generated* $\mathbf{y}$ *images from the expected hidden activations.*

|  | Linear | C-GRBM +Linear | SFNN +Linear | MLP | SFNN +MLP |
|---|---|---|---|---|---|
| clean | 80.0% | 81.4% | **82.4%** | 83.2% | 83.8 % |
| 10% noise | 78.9% | 79.7% | 80.8% | 82.0% | 81.7 % |
| 50% noise | 72.4% | **74.3%** | 71.8% | 79.1% | 78.5% |
| 75% noise | 52.6% | 58.1% | **59.8%** | 71.9% | **73.1%** |
| 10% occl. | 76.2% | 79.5% | 80.1% | 80.3% | **81.5%** |
| 50% occl. | 54.1% | 59.9% | **62.5%** | 58.5% | **63.4%** |
| 75% occl. | 28.2% | 33.9% | **37.5%** | 33.2% | **39.2%** |

Table 3: *Recognition accuracy over 5 folds. Bold numbers indicate that the difference in accuracy is statistically significant than the competitor models, for both linear and nonlinear classifiers.*

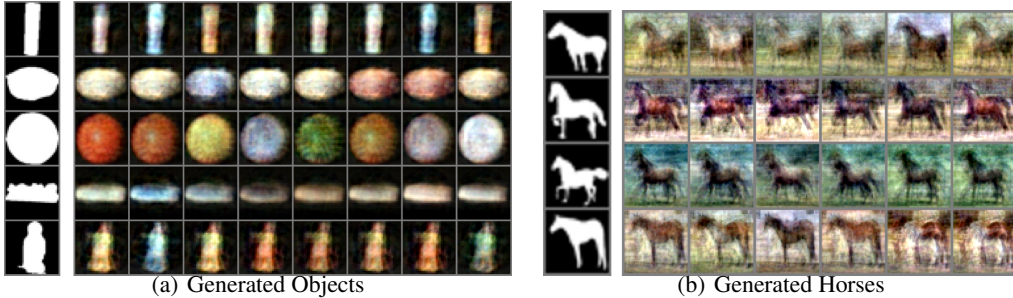

<div align="center">(a) Generated Objects        (b) Generated Horses</div>

Figure 7: *Samples generated from a SFNN after training on object and horse databases. Conditioned on a given foreground mask, the appearance is multimodal (different color and texture). Best viewed in color.*

SFNNs are also useful when dealing with noise. As a generative model of $\mathbf{y}$, it is somewhat robust to noisy and occluded pixels. For example, the left panels of Fig. 6, show corrupted test images $\mathbf{y}$. Using the importance sampler described in Sec. 2.1, we can compute the expected values of the binary stochastic hidden variables given the corrupted test $\mathbf{y}$ images[5]. In the right panels of Fig. 6, we show the corresponding generated $\mathbf{y}$ from the inferred average hidden states. After this denoising process, we can then feed the denoised $\mathbf{y}$ and $E[\mathbf{h}]$ to the classifiers. This compares favorably to simply filling in the missing pixels with the average of that pixel from the training set. Classification accuracies under noise are also presented in Table 3. For example 10% noise means that 10 percent of the pixels of *both* $\mathbf{x}$ and $\mathbf{y}$ are corrupted, selected randomly. 50% occlusion means that a square block with 50% of the original area is randomly positioned in both $\mathbf{x}$ and $\mathbf{y}$. Gains in recognition performance from using SFNN are particularly pronounced when dealing with large amounts of random noise and occlusions.

### 3.3 Additional Qualitative Experiments

Not only are SFNNs capable of modeling facial expressions of aligned face images, they can also model complex real-valued conditional distributions. Here, we present some qualitative samples drawn from SFNNs trained on more complicated distributions (an additional example on rotated faces is presented in the Supplementary Materials).

We trained SFNNs to generate colorful images of common objects from the Amsterdam Library of Objects database [20], conditioned on the foreground masks. This is a database of 1000 everyday objects under various lighting, rotations, and viewpoints. Every object also comes with a foreground segmentation mask. For every object, we selected the image under frontal lighting without any rotations, and trained a SFNN conditioned on the foreground mask. Our goal is to model the appearance (color and texture) of these objects. Of the 1000 objects, there are many objects with similar foreground masks (e.g. round or rectangular). Conditioned on the test foreground masks, Fig. 7(a) shows random samples from the learned SFNN model. We also tested on the Weizmann segmentation database [21] of horses, learning a conditional distribution of horse appearances conditioned on the segmentation mask. The results are shown in Fig. 7(b).

## 4 Discussions

In this paper we introduced a novel model with hybrid stochastic and deterministic hidden nodes. We have also proposed an efficient learning algorithm that allows us to learn rich multi-modal conditional distributions, supported by quantitative and qualitative empirical results. The major drawback of SFNNs is that inference is not trivial and $M$ samples are needed for the importance sampler. While this is sufficiently fast for our experiments we can potentially accelerate inference by learning a separate recognition network to perform inference in one feedforward pass. These techniques have previously been used by [22, 23] with success.

## Footnotes

[1] For example, in a MLP with one input, one output and one hidden layer: $p(y|\mathbf{x}) \sim \mathcal{N}(y|\mu_y, \sigma_y^2)$, $\mu_y = \sigma\big(W_2\sigma(W_1\mathbf{x})\big)$, $\sigma(a) = 1/(1 + \exp(-a))$ is the sigmoid function. Note that the Mixture of Density Network is an exception to the unimodal assumption [1].

[2] An equivalent problem is learning one-to-many functions from $X \mapsto Y$.

[3]In practice, due to weight sharing, we will not be able to have close to that many modes for a large $N_h$.

[4]It is straightforward to extend the model to multiple and hybrid hidden layered SFNNs.

[5]For this task we assume that we have knowledge of which pixels is corrupted.

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
