[Supplementary Material]

# Supplementary Materials for
# Learning Stochastic Feedforward Neural Networks

**Yichuan Tang**
Department of Computer Science
University of Toronto
Toronto, Ontario, Canada.
tang@cs.toronto.edu

**Ruslan Salakhutdinov**
Department of Computer Science
University of Toronto
Toronto, Ontario, Canada.
rsalakhu@cs.toronto.edu

## 1   Learning Algorithm

---
**Algorithm 1** EM learning algorithm

---
Given training $D$ dimensional data pairs: $\{\mathbf{x}^{(n)}, \mathbf{y}^{(n)}\}$, $n = 1 \ldots N$. Hidden layers $\mathbf{h}^1 \& \mathbf{h}^4$ are deterministic, $\mathbf{h}^2 \& \mathbf{h}^3$ are hybrid. $\theta = \{W^{1,2,3,4,5}, bias, \sigma_y^2\}$
**repeat**
  *//Approximate E-step*:
1  Compute $p(\mathbf{h}^2|\mathbf{x}^{(n)}) = Bernoulli(\sigma(W^2 \sigma(W^1 \mathbf{x}^{(n)})))$
2  $\mathbf{h}^2_{determ} \leftarrow p(\mathbf{h}^2_{determ}|\mathbf{x}^{(n)})$
  **for** $m = 1$ **to** $M$ (importance samples) **do**
3    Sample: $\mathbf{h}^2_{stoch} \sim p(\mathbf{h}^2_{stoch}|\mathbf{x}^{(n)})$.
    let $\mathbf{h}^2$ be the concatenation of $\mathbf{h}^2_{stoch}$ and $\mathbf{h}^2_{determ}$.
4    $p(\mathbf{h}^3|\mathbf{x}^{(n)}) = Bernoulli(\sigma(W^3 \mathbf{h}^2))$
5    $\mathbf{h}^3_{determ} \leftarrow p(\mathbf{h}^3_{determ}|\mathbf{x}^{(n)})$
6    Sample: $\mathbf{h}^3_{stoch} \sim p(\mathbf{h}^3_{stoch}|\mathbf{x}^{(n)})$
    let $\mathbf{h}^3$ be the concatenation of $\mathbf{h}^3_{stoch}$ and $\mathbf{h}^3_{determ}$.
7    Compute $p(\mathbf{y}|\mathbf{x}^{(n)}) = \mathcal{N}(\sigma(W^5 \sigma(W^4 \mathbf{h}^3)); \sigma_y^2)$
  **end for**
8  Compute $w^{(m)}$ for all $m$, using Eq.5.

  *//M-step*:
  $\triangle \theta \leftarrow 0$
  **for** $m = 1$ **to** $M$ **do**
9    Compute $\frac{\partial Q^{(m)}(\theta, \theta_{old})}{\partial \theta}$ by Backprop.
10    $\triangle \theta = \triangle \theta + \partial Q^{(m)}/\partial \theta$
  **end for**
11  $\theta_{new} = \theta_{old} + \frac{\alpha}{M} \triangle \theta$, *//$\alpha$ is the learning rate.*
**until** convergence

---

## 2   Additional Qualitative Experiments

Not only SFNNs are capable of modeling facial expression of aligned face images, they can also model complex real-valued conditional distributions. In this section, we present some qualitative samples drawn from SFNNs trained on more complicated distributions. We tested the UMIST faces database [1], which contains in-depth 3D rotation of heads.

We trained the same SFNNs as described in the previous section on the 16 training subjects, using 4 subjects for testing. Conditioned on the profile view, we are modeling the distribution of rotations up to 90 degrees. Fig. 1 displays 3 test subjects' profile view along with seven exact samples drawn

Figure 1: *Samples from SFNN trained on rotated faces.*

from the model (plotted on the right hand side). This is a particularly difficult task, as the model must learn to generate face parts, such as eyes and nose.

## 3   Computation Time

Despite having to draw $M$ samples during learning, Fig.5 empirically demonstrated that 20 samples is often sufficient[1]. This is in part due to the fact that samples from the conditional prior are exact and in part due to the cooperation that occurs during learning (Sec.2.2). Regarding hardware, our experiments are performed on nVidia GTX580 GPUs. This gives us over 10x speedup over CPUs. For example, a 4 hidden layer SFNN with 2304 input and output dimensions, 128 stochastic hidden nodes, and 50 samples per E-step, can update its parameters in 0.15 secs on a minibatch of 100 cases.

In Table 2, C-GRBM is also trained on the GPU, but is much slower due to its use of a large hidden layer and 25 CD steps. For example, the C-GRBM requires 1.16 secs per parameter update. MFA and MDNs are run on CPUs and we can also expect 10x speedup from moving to GPUs.

## Footnotes

[1]We note that this is still $M$ times more expensive than standard backprop.

## References

[1] Daniel B Graham and Nigel M Allinson. Characterizing virtual eigensignatures for general purpose face recognition. In *Face Recognition: From Theory to Applications*, pages 446–456. 1998.