[Reviews · NeurIPS 2013]

Submitted by Assigned_Reviewer_5

This paper proposes a neural network architecture that falls somewhere between multilayer perceptrons (MLPs) and sigmoid belief networks (SBNs). The motivation is to permit multimodal predictive distributions (like SBNs) by using stochastic hidden units, but adds deterministic hidden units to smooth the predictive distribution in the case of real-valued data. The paper's main technical contribution is an EM-style algorithm where the E-step uses importance sampling to approximate the posterior and the M-step uses backpropagation to update the parameters. The experiments demonstrate the model's utility on several synthetic and real datasets.

Quality: I liked this paper; the use of stochastic and deterministic units seems reasonably justified. The model and inference/learning were well presented and I thought the experiments were thorough. One criticism would be that the datasets considered are either small or subsets of other better-known datasets. In other words, large ML baseline datasets are not considered. However, the paper makes an effort to compare the proposed model against a number of reasonable baselines.

Clarity: The paper was generally well organized and well-written. It has a number of small grammatical errors and would benefit from a thorough proofread.

Originality: I am not aware of other feedforward neural network architectures that use a mix of deterministic and stochastic units. The inference and learning procedure, while approximate, seem reasonable (and bear a slight resemblance to the sampling methods used in RBM-type models).

Significance: The paper is correct in that there is often a need to model multimodal predictive distributions in machine learning. The proposed model is relatively easy to understand and should be able to be easily implemented and extended. I think it would be considered by other practitioners.

Comments

Figure 1 presents a model where the first and last layer are wholly deterministic and the inner layers contain some stochastic units. Once the reader gets to the experiments, one sees that is the type of architecture used in the experiments. But the paper does not elaborate on this choice. Is there some significance in choosing a first and last fully-deterministic layer, or is this arbitrary?

Is it necessary to use sigmoid units for both the deterministic and stochastic units? Other types of nonlinearities are not mentioned in the paper.

Page 2, "We note that we can use any other parameterized distribution ... for the output variables. This is a win compared to the Boltzmann Machine family of models, which require the output distribution to be from the exponential family." Is this totally true? I agree that exponential family Boltzmann machine models are convenient and the norm, but isn't one completely free to define the energy function and interpret the distributions however one chooses?

The baseline SBN considered in the experiments is trained in the exact same way as the SFNN, only it has no deterministic units. Have you compared to Sigmoid Belief networks trained in other ways? Is there an advantage to training them with the proposed algorithm designed for SFNNs? That would be an interesting secondary result.

The experiments in section 3.1 do not say how the test log-probability was approximated for the C-GRBM (which has an intractable partition function, so log likelihood cannot be computed exactly). The following section says that AIS was used; is this also the case for section 3.1?

Figure 5 shows how different hyperparamters affect the evaluation and learning of SFNNs, including the absolute number of stochastic nodes per layer. I'd be interested in seeing the ratio of stochastic to deterministic nodes, rather than the absolute number of stochastic nodes.


Minor comments
- Footnote of pg 1, \sigma is overloaded (used for both variance e.g. \sima_y^2 and also to mean the sigmoid function)
- Inline math immediately after Eq 1, bias_2, bias_1 is ugly notation; why not use a lowercase letter for the biases?
Summary: The paper presents an interesting feedforward architecture that makes use of deterministic and stochastic hidden units. Inference and learning are approximate but appear to be practical and effective on a number of datasets.

Submitted by Assigned_Reviewer_6

The paper proposes a multi-layer perceptron (architecture and learning algorithm) capable of modeling multi-modal output distributions. The authors achieve this goal with an idea by Neal (1992), by interpreting the network output as a probability distribution with stochastic hidden-layer neurons, which can be sampled from. Improving on the earlier work, the authors propose an efficient sampling algorithm using importance sampling and a approximate EM training algorithm which makes use of this sampling procedure. Learning and learned models are analyzed in depth, using synthetic and real datasets. The model seems to be capable of performing tasks typically left to the much more complicated-to-learn restricted Boltzmann machine family of algorithms in the deep learning community (completion of missing inputs, sampling from a modelled distribution).

The paper has very good quality. Theory as well as experimental setup and results are well-presented (although this reviewer would prefer the training algorithm to be part of the paper, not the supplement). In the experiments, the proposed machinery is compared with a conditional Gaussian RBM. The interactions between sampled features in this model are quite limited, wouldn't a stacked CRBM version be more appropriate as a comparison with your multi-layered model?

As far as this reviewer can tell, the material is novel and a significant contribution.

minor: the indefinite article before "MLP" should be "an", not "a" (it's a phonetic rule IIRC).
Summary: The paper proposes a multi-layer perceptron (architecture and learning algorithm) capable of modeling multi-modal output distributions. The paper is of high quality, the proposed algorithm improves state-of-the-art and is analyzed thoroughly.

Submitted by Assigned_Reviewer_9

This paper introduces a feed forward neural network that uses a combination of stochastic binary units and deterministic units to reconstruct the input signal. By drawing several samples of the stochastic states, the network can learn multimodal distributions of the input data. The paper shows applications on synthetic data and facial expression modeling as well as shows samples drawn from the model when a silhouette of the input object is used. The paper seems clearly presented though the toy dataset are somewhat limiting and therefore may not have as significant impact as possible.

Pros
Good mixing when sampling different modes of facial expression based on the mean face expression

Interesting generative outputs given only the segmentation masks of horses and other objects.

Cons
For classification the representations needed to be concatenated with the pixels. There should have been results reported with the representation only as well.

Training is slower than a traditional feed forward net due to the extra sampling which is also needed, even in higher quantities, at test time.

Perhaps Boltzmann machines which learn a joint distribution of all layers (much like the backprop in this paper) would be a better comparison than crbms.

The training procedure is much slower than MFA for only a marginal improvement in log probs. This is made worse by the fact that the proposed algorithm is trained on a GPU which is significantly faster.

Comments:
Taking the input as the mean image might not be optimal. One could imagine a scenario where any input image is used to predict any output image. This might tease apart the different modes even more during training.

What happens to the deterministic units when the stochastic ones are sampled M times? Are they averaged M times as well? For the first layer deterministic activations this would not make a difference as they have the same value always. But for a second layer of them, that take as input the stochastic units in layer 1, these could vary significantly each of the M times.

Can this be trained like dropout where only 1 sample is drawn each step? Is there a fast 1 step inference?

It's hard to know why the deterministic units wouldn't just dominate and take over all the modeling capacity. If there is an easy path of a network it usually chooses that one, and adding noise on one path is not going to make that path easy.
Summary: The approach presented in the paper is an interesting way to help a feedforward net tease apart different modes in the input data. However, the somewhat toy experiments don't do the method justice.
Author Feedback

Author rebuttal: Reviewer 5:
It is not a critical requirement to set H1 and H4 to be fully deterministic. We decided to make them deterministic because we wanted to use a nonlinear function to determine the probabilities of the stochastic H2 nodes. We also wanted the stochastic H3 nodes to activate the output Y nonlinearly.

Sigmoid units were chosen due to the connection with traditional MLP and Conditional RBMs. Other types of nonlinearities for both the stochastic (e.g. discrete/multinomial) and deterministic nodes can indeed be used and could perform better, given the recent success of deep learning with rectified linear units.

Boltzmann Machines have a bilinear energy term and define a probability over states using the Gibbs distribution. There are other models such as Product of Student't which can model distribution not from the exponential family, but those models have disadvantages compared to models like restricted Boltzmann machines (such as not having exact inference for their latent variables).

We have tried to train SBNs with Gibbs sampling and variational methods. Both were extremely slow as they had to consider each latent variables one at a time. In addition, mean-field approximation is not very suitable for continuous data as it requires additional approximations making the lower-bound even looser.

AIS is used for section 3.1 as well.

Reviewer 6:
Stacked conditional RBMs, or equivalently, conditional deep Boltzmann machines are indeed a better comparison. It was hard to efficiently learning a conditional DBM due to the sampling requirements of the negative phase.

A quantitative evaluation of a DBM is also tricky as not only AIS must be used to approximate the partition function, but it is also intractable to sum-out all of the latent variables given a particular data x, which is required in order to compute the log probability densities.

Reviewer 9:
Every time the stochastic nodes are sampled, it leads to the corresponding changes to their connected deterministic units in the next layer. For example, in Fig. 1, sampling stochastic nodes of H2 would not change the deterministic nodes of H2, but it will change the deterministic part of H3, as they are connected to the stochastic nodes of H2. The activations of the layers are not averaged, but rather the gradients of the weights for each sampling run are averaged (see Eq. 6).

There might be a 1-sample approximation similar to stochastic approximation procedure, we are looking into that for future work. However, our current algorithm requires the correct importance weights w^(m), which must be properly estimated.